# Effects of Irrigation Discharge on Salinity of a Large Freshwater Lake: A Case Study in Chagan Lake, Northeast China

**Xuemei Liu [1,2], Guangxin Zhang [1,*], Jingjie Zhang [3], Y. Jun Xu [4], Yao Wu [1], Yanfeng Wu [1], Guangzhi Sun [1], Yueqing Chen [1] and Hongbo Ma [5]**

[1]  Northeast Institute of Geography and Agroecology, Chinese Academy of Sciences, Changchun 130102, China; liuxuemei@iga.ac.cn (X.L.); wuyao@iga.ac.cn (Y.W.); wuyanfeng@iga.ac.cn (Y.W.); bigbang.theorist@yahoo.com (G.S.); chenyueqing@iga.ac.cn (Y.C.)

[2]  University of the Chinese Academy of Sciences, Beijing 100049, China

[3]  Department of Civil & Environmental Engineering, National University of Singapore, Kent Ridge 117576, Singapore; lakejz@gmail.com

[4]  School of Renewable Natural Resources, Louisiana State University Agricultural Center, 227 Highland Road, Baton Rouge, LA 70803, USA; yjxu@lsu.edu

[5]  Jilin Meteorological Service, Changchun 130062, China; chenliwen86@126.com

*  Correspondence: zhgx@iga.ac.cn; Tel.: + 86-0431-130-8941-0676; Fax: + 86-0431-8554-2298

**Abstract:** The salinization of freshwater lakes by agricultural activities poses a threat to many lake ecosystems around the world. Quantitative, medium- to long-term studies are needed to understand how some common agricultural practices, such as the discharge of crop irrigation in the vicinities of large lakes, may affect lake salinization. In this study, hydrological, hydrodynamics, water quality and meteorological datasets were used to analyze the long-term spatial-temporal variations of water salinities of a major lake, the Chagan Lake, in Northeast China. An integrated hydrodynamics-salinity model was used to simulate lake water salinity changes taking place at different times and locations, including (i) salt accumulations during a non-frozen period, and (ii) the time when water salinity may reach a significant threshold (1 psu) that jeopardizes a major environmental and economic value of this lake (i.e., the cultivation of local fish species). The results confirmed that Chagan Lake was indeed undergoing salinization in the ten year period between 2008 and 2018. The spatial-temporal patterns of the salinization processes were identified. For instance, (i) the mean salinity of the lake water was found to be 0.55 psu in the summer season of the region and 0.53 psu in the winter, and (ii) between May to October the salinity was up to 0.62 psu in the western region of the lake. The rate of salt accumulation was found to be 97 ton per annum during the non-frozen period. The simulation predicted that by 2024 the lake water will become sub-saline (salinity > 1.07 psu) which is toxic to fish species, if the current practice of irrigation discharge into the lake continues. In the scenario that the amount of irrigation discharges into the lake doubles, the western region of the lake will become sub-saline within one year, and then the whole lake within three years. Overall, this study has produced results that are useful to authorities around the world, for balancing the risks and benefits of developing crop irrigation fields in areas surrounding large freshwater lakes.

**Keywords:** lake salinization; irrigation discharge; hydrodynamics-salinity modeling; Chagan Lake

## 1. Introduction

The salinization of freshwater has received increasing attention because it can damage the ecosystem, mainly biological communities [1–5]. Although lake water salinities may vary

cyclically with climatic processes [6], significant salinization in the past decades has been largely due to anthropogenic activities, such as agricultural production [7–10], industrial sewage [11], human-accelerated weathering [12,13], and land clearing [14,15]. In arid and semi-arid regions, agricultural activities are the dominant factors of lake salinization [16]. Direct discharges of crop irrigation waters can drastically change the salinity of freshwater lakes [17]. Systematic studies on how agricultural activities can affect lake salinization are becoming increasingly urgent as the sources of freshwater become scarcer.

A number of studies have shown that agricultural activities—which use pesticides and fertilizers, groundwater extraction for irrigation, unreasonable irrigation and drainage methods—have accelerated the accumulation of salt, causing soil salinization [18–20] and groundwater salinization [21–23]. These are believed to eventually bring about reductions in cropping productivities [24,25]. These studies focused on the effects of agricultural activities on soil salinization and water body salinization on irrigation areas. It is also known that irrigation discharge would increase lake salinity. However, these studies rarely address the effect of irrigation discharge on discharge areas (the area receiving irrigation discharge), especially long-term simulation and prediction. Previous studies also shown that intensified irrigation development can cause ecological degradation in areas surrounding freshwater lakes [26–30], although the degradations may be buffered by the presence of wetlands in some areas [31]. Ficker et al. [32] identified that in cases of strongly reduced or ceased salt pollution of lakes, water renewal could change water density stratification, resulting in reduced oxygen levels in the bottom of deep lakes.

Most previous studies on lake salinization used $Cl^-$, total dissolved solids (TDS), or electrical conductivity (EC) rather than salinity to measure how irrigation discharge affects freshwater lake salinity [33–35]. Some researchers demonstrated that $Cl^-$ and TDS cannot replicate inherent variation of salinity in freshwater lakes, especially in inland waters [36,37]. Therefore, further investigation is needed to see how and to what extent freshwater lake salinity has changed and will evolve when considering irrigation areas.

In general, studies on lake salinity are either field-based [38,39] and focused on a static study modeling the watershed outlet [40], or model-based which involves complex modelling at a plot scale [41–43]. With the unceasing promotion of computer performance, numerical modeling integrated with field and laboratory data has proven to be an effective tool for describing the complex physical [44–46], chemical, and biological processes [47,48] and their interactions between state variables [49,50]. Since numerical modeling can replicate variation of salinity at any spatial-temporal scale, extensive studies have been done to explore the effects of agricultural activities on lake salinity [6,20,51,52]. Moreover, more modeling is necessary to understand the vulnerability of salinity in various ecosystems [53], to subsequently aid in the formulation of sustainable water management [54] and development plans in irrigation areas [55]. However, little has been done to simulate the impact of irrigation discharge on long-term variations of salinity in inland freshwater lakes that receive agricultural discharge with high salinity.

The majority of China's saline lakes (salinity > 3 psu) [56] are located in Northwest China [57]. However, freshwater lakes have a high risk of salinization in this region due to further development of large-scale centralized agriculture. Located in Northeast China, Chagan Lake is among the ten largest freshwater lakes in China in terms of area, and it is an important base of freshwater fisheries [58]. The lake was once surrounded by natural sodic saline-alkali grassland half a century ago [59]. Its drainage area gradually converted into one of China's rice production centers primarily during the 1970s and 1980s [60]. Since then, the lake has been continuously receiving irrigation discharge from over $8.73 \times 10^4$ hectares of rice paddies. The continuous discharge has increased the risk of water salinization due to a growing amount of discharge from irrigation regions, thus threatening lake water quality and ecosystem health.

In this study, we analyzed historical datasets collected from monitoring stations and simulated spatial-temporal changes of lake salinity in Chagan Lake with the aim to: (i) reveal the long-term

variations and spatial-temporal distribution of salinities of water in this lake; (ii) estimate salt accumulation during the non-frozen period; and (iii) predict how long it would take to reach the salinity threshold (1psu) for freshwater fish farming using integrated hydrodynamics and salinity modeling.

## 2. Materials and Methods

### 2.1. Study Area

Located on the Songnen Plain, Northeast China (124°03′–124°34′ E, 45°09′–45°30′ N; Figure 1), Chagan Lake covers a surface area of 372 km$^2$ and has an average depth of 2.5 m. The regional climate is semi-arid and sub-humid continental monsoon, with a long-term annual average temperature of 5.5 °C and a long-term annual average evaporation and precipitation of 1449 mm and 430 mm, respectively.

From the 1960s to the present, precipitation, groundwater, and irrigation discharge have been the main water source inputs into Chagan Lake. In recent years, irrigation discharges that account for 64% [58] of the inputs have caused increasing concerns over further salinization. The lake is surrounded by three irrigation districts, including an old irrigation district (Qianguo district) and two new irrigation districts (Da'an and Qian'an districts) covering $5.07 \times 10^4$, $2.19 \times 10^4$ and $1.33 \times 10^4$ hectares, respectively (Figure 1). As the lake receives discharge from the Qianguo and Da'an irrigation districts, nutrients, salt, and alkali flowing into the lake have caused eutrophication and salinity to increase [61,62]. According to the development plan of the Songyuan irrigation district, 'the irrigation discharge from the Qian'an irrigation district will flow into Chagan Lake in the future. Some physical and chemical characteristics of the lake water are presented in Table 1.

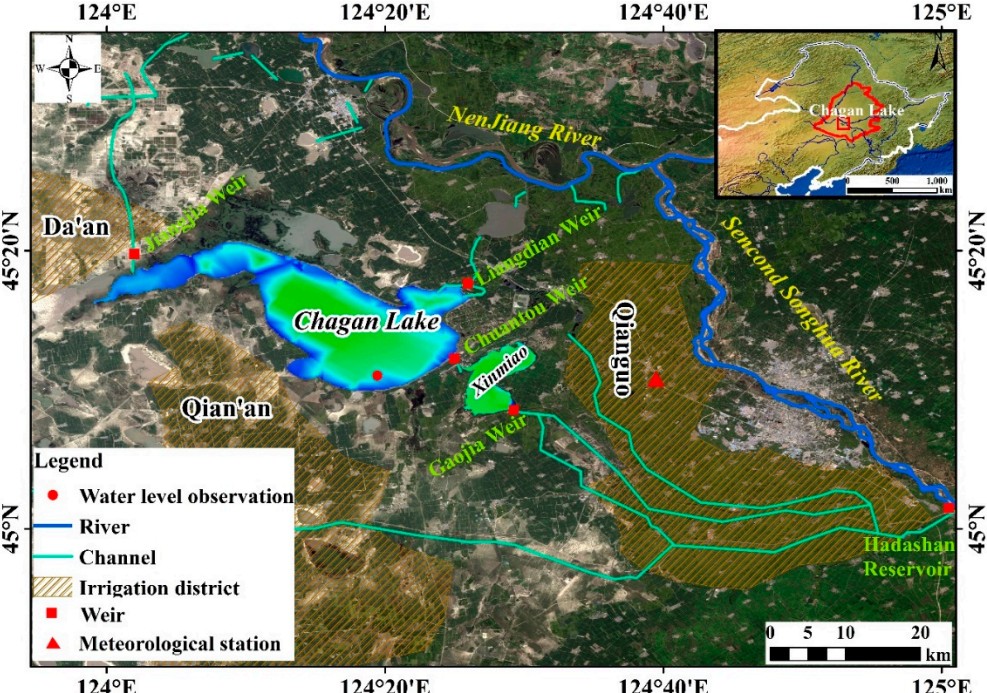

**Figure 1.** Location of the study area.

**Table 1.** Physical and chemical characteristics of Chagan Lake.

| Parameters | Mean | Range |
|---|---|---|
| Wind (m/s) | 2.19[a] | 0–26.6 |
| Water Depth (m) | 2.50[b] | 0.5–4.5 |
| Water temperature (°C) | 15.48[a] | 3.5–30.20 |
| Air temperature (°C) | 3.93[a] | −28.2–33.9 |
| Salinity (psu) | 0.49[b] | 0.31–0.78 |
| pH | 8.75[b] | 8.20–9.13 |
| Electrical conductivity (μs/cm) | 0.94[b] | 0.49–1.07 |
| Precipitation (mm/h) | 0.08[a] | 0–28.4 |
| Evaporation (mm/h) | 0.12[a] | 0–14.7 |
| Relative humidity (%) | 70.38[a] | 7–99 |
| Solar radiation (w m$^2$) | 263.00[a] | 0–400.04 |

Letters [a] and [b] represent significance levels of parameters passing the 0.0001 and 0.001 significance test, respectively. The hourly meteorological datasets including wind, water temperature, air temperature, precipitation, evaporation, relative humidity and solar radiation were derived from the Qianguo Meteorological Station (http://data.cma.cn). The water quality data are an average value from May to October in 2018.

## 2.2. Data Collection and Model Inputs

### 2.2.1. Sample Collections

A large quantity of data including sets of hydrological, hydrodynamics, water quality and meteorological data were collected from field samples and data acquisitions from monitoring stations during the non-frozen period (May to October 2018). Hydrological datasets including irrigation discharges, water depth, and flow velocity were monitored using a real-time monitoring system at the main inlet and outlet of Chagan Lake established in 2018. Field investigations started when water samples were collected from nine monitoring stations (S1–S6; A1–A3), as shown in Figure 2, from May to October 2018. Among the monitoring stations, A1–A3 are adjacent to the Jiangjia station, Liangdian station, and Chuantou station. Station (S1) is in the western region of Chagan Lake, station S2, S3, and S4 are in the middle region of Chagan Lake, and station S5 and S6 are in the eastern region. Immediately after water sample collection, all the water samples were analyzed in the testing center of the Northeast Geographic and Agricultural Research Institute of the Chinese Academy of Sciences. The values of $Cl^-$ and $SO_4^{2-}$ were analyzed using an ion chromatograph (IE-009). $Mg^{2+}$, $Ca^{2+}$, $Na^+$, and $K^+$ were analyzed using an atomic absorption spectrophotometer (IE-001). $HCO_3^-$ and $CO_3^{2-}$ were analyzed by acidometer titration method. Additional salinity data were collected from the literature [62–66] and previous field investigations.

The setting of salinity threshold (1 psu) was based on the Fishery Industry Standard of the People's Republic of China (2008). Water temperature and pH in the lake (station A1–A3; S1–S6) were collected monthly between May to October 2018 using a HANNA measurement probe.

### 2.2.2. Model Inputs

The meteorological datasets including hourly wind speed, wind direction, precipitation, evaporation, solar radiation, cloud cover, relative humidity and air temperature, etc. in 2018 were derived from the Qianguo meteorological station (http://data.cma.cn/) (Table 2). Parameters such as roughness, Dalton, viscosity and Stanton number used in the hydrodynamics model were obtained from the literatures [67,68] (Table 2). These datasets of salinity, temperature and water levels between May and October were used for model calibration and validation, respectively. In addition, geometric data including bathymetry and land boundaries were obtained based on Google Maps. The tools of RGFGRID and QUICKIN provided by Delft3D software suite have been used to create grids and incorporate bathymetry data (Figure 2). A grid of finite difference quadrangular elements was created for the whole Chagan Lake with 4928 cells as shown in Figure 2.

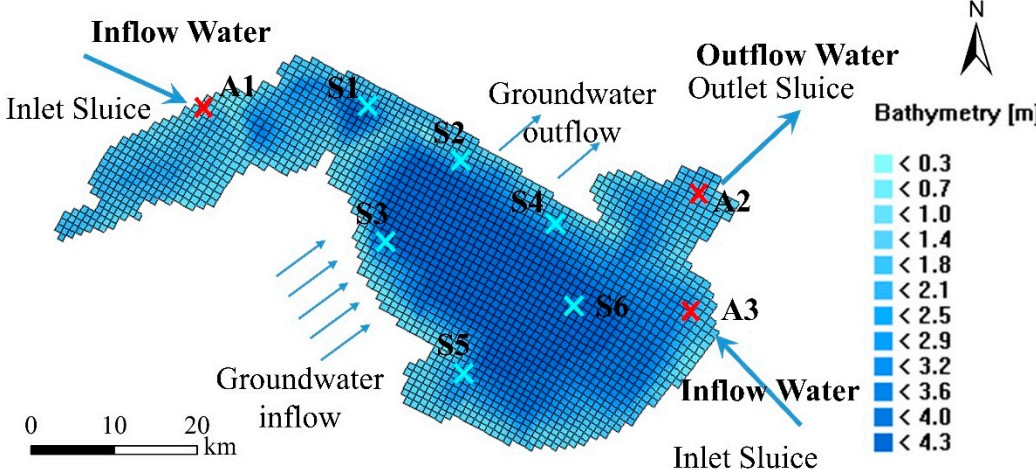

**Figure 2.** Sampling stations and bathymetry of Chagan Lake. Squares indicate the calculation cell.

**Table 2.** Physical parameters and coefficients used in the integrated hydrodynamics-salinity model.

| Parameters | Value | Use Category | Unit | Source |
|---|---|---|---|---|
| Gravity | 9.81 | Constants | m s$^{-2}$ | |
| Water density | 1000 | Constants | kg/m$^3$ | Lab value |
| Air density | 1.0 | Constants | kg/m$^3$ | |
| Wind drag coefficients at wind speed, range of 0–6 m/s | 0.00063 | Wind stress | - | [67] |
| Wind drag coefficients at wind speed, range of 6–26 m/s | 0.00723 | Wind stress | - | [67] |
| Wind drag coefficients at wind speed above 26 m/s | 0.00723 | Wind stress | - | [67] |
| Bed roughness for Manning Roughness formulation | 0.022 | Roughness | - | [68] |
| Background eddy viscosity | 2.0 | Viscosity | m$^2$/s | Dependent on grid size |
| Background eddy diffusivity | 2.0 | Viscosity | m$^2$/s | Dependent on grid size |
| Secchi depth | 0.5 | Heat flux model | m | Field survey |
| Dalton number for evaporative heat flux | 0.0013 | Heat flux model | - | [69] |
| Stanton number for heat convection | 0.0013 | Heat flux model | - | [69] |

## 2.3. Model Setup, Calibration and Simulation

### 2.3.1. Modeling Setup

An integrated hydrodynamics-salinity model was built based on the Delft3D modelling suite (Figure 3) to simulate unsteady flow, transport phenomena and water temperature of Chagan Lake. The model also provided the hydrodynamics basis for the salinity simulation. Since the model was based on the orthogonal curvilinear coordinate system—which included a horizontal moving solution, a continuous equation, and a transmission equation of anti-corrosion components [70]—the flow was affected by the law of conservation (mass, momentum, and energy). In shallow water and based on the Boussinesq hypothesis, the Navier-Stokes equation was used in the hydrodynamics-salinity model for incompressible fluid [71,72]. If the flow was turbulent, the system also incorporated other turbulent transport equations that were mathematical descriptions of these constant laws. In this study, the influence of water temperature on water density was neglected, and the water density of the whole flow field was considered faithful. At the same time, it was assumed that the pressure in the vertical direction was hydrostatic pressure distribution.

The observations of water level, salinity, and temperature in May 2018 were first used as initial conditions. In order to stabilize the model, the spin-up time of the model was four months (from January to April) with the time step being 1 minute. Then, the initial conditions were reset with a restart file from this warm-up run. The model stability was also checked with the continuity check suggested by the delft3D user manual with a constant value of 1.0 over the simulation period.

Given the lake is very shallow with an average depth of less than 2.5 m, it was assumed the lake was well-mixed with no vertical stratification, and a 2D hydrodynamics model was applied. The simulation period was 6 months between May and October, whereas the frozen-period between

November and April was assumed to have no influence on the results (no inflow and outflow into or out of the lake during this period).

As shown in Figure 3, integrated modeling primarily consisted of four major blocks: (i) the input dataset including meteorological data, discharge data, and salinity loading; (ii) integrated process-based modeling; and (iii) salinity simulation and prediction. The initial conditions were read from a restart file from a previous run. According to the heat balance between oceanography and the atmospheric system, the total heat flux of the free surface was obtained to simulate salinity variations in the lake.

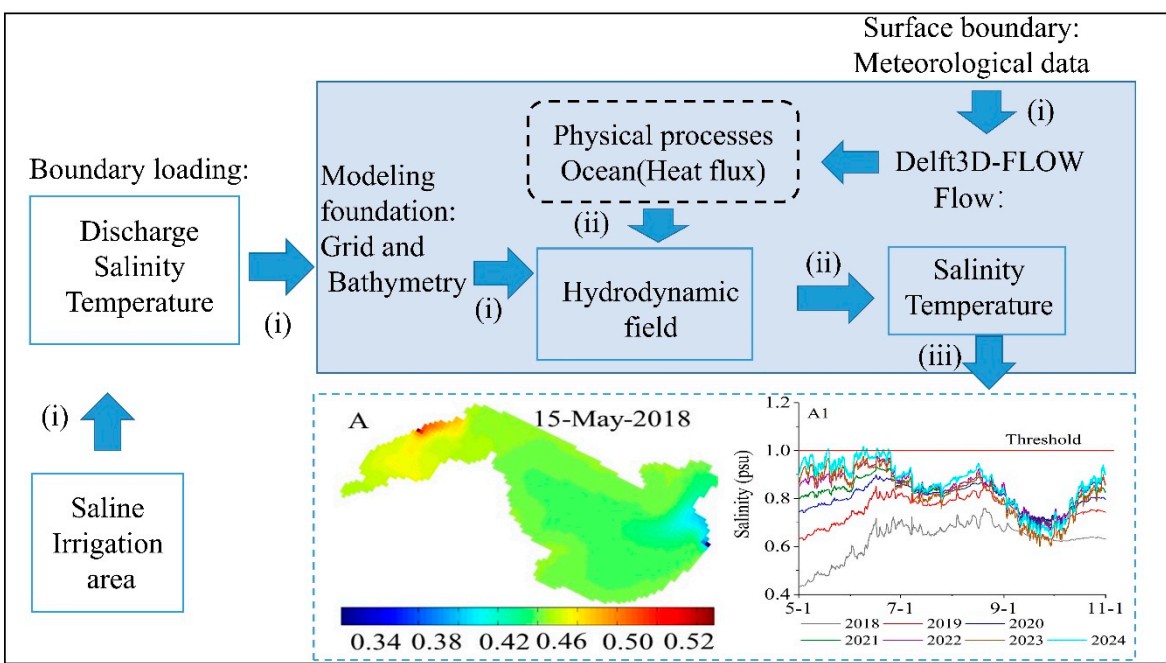

**Figure 3.** Conceptual diagram of integrated modeling framework, where (i), (ii), and (iii) indicate the major regions of the modeling framework.

### 2.3.2. Boundary and Discharge

A grid of finite difference quadrangular elements was created for the whole of Chagan Lake with 4928 cells as shown in Figure 2. On the closed boundary of Chagan Lake, water discharge was zero. The flow inside the lake was formed by the discharge at the open boundaries, wind stress at the free surface, and pressure gradients. The wind and water flow were the main driving forces of salt movement in the lake. The inflow into the lake included water supply from the inlet and groundwater interchange and outflows included discharge from the outlet and interchange with the groundwater. The warm-up period of the model was set from January to April 2018 in order to stabilize the operation of the model. Water quantity values of boundaries in 2018 are presented in Table 3.

**Table 3.** Water balance of inflow and outflow of Chagan Lake in 2018.

|  | Inflow Water ($10^8$ m$^3$) | | | | Outflow Water ($10^8$ m$^3$) | | |
|---|---|---|---|---|---|---|---|
| Boundaries | P | CT | JJ | $G_{in}$ | E | LD | $G_{out}$ |
| Water quantity | 1.18 | 1.60 | 0.91 | 0.90 | 2.81 | 1.54 | 0.24 |
| Sources | a | b | b | c | a | b | c |

Letters a and b refer to Qianguo meteorological and yield observation, respectively; c refers to data from Zhang et al. [73]; P, E, CT, JJ, $G_{in}$, LD, and $G_{out}$ indicate precipitation, evaporation, inflow waters from Chuantou, Jiangjia, the groundwater, and outflows from Liangdian and the groundwater, respectively.

### 2.3.3. Calibration and Validation

Model calibration and validation were conducted by comparing simulated results with the observations [53,74] from May to October 2018. Given the high uncertainty in the limited dataset, the calibration was done mainly through adjusting inflow and outflow rates. The parameters such as roughness, Dalton, viscosity and Stanton number obtained from the literature were slightly adjusted as presented in Table 2. Figure 4 illustrates the comparison between simulated and measured water depths at the A1 station. The simulated results showed very good agreement with the measured water depths. Simultaneously, in order to increase the accuracy of the model, salinity and temperature were further calibrated using monthly observation data (S1–S6). To assess the performance of the model, the Nash–Sutcliffe efficiency (NSE) model as standard regression and percent bias (PBIAS) as error index were used [75]. The performance of the model was considered to be satisfactory when NSE was above 0.36 and PBIAS was below 25 [76,77]. The calculations of NSE and PBIAS are shown in the following equations:

$$\text{NSE} = 1 - \left[ \frac{\sum_{i=1}^{n}\left(x_i^{\text{obs}} - x_i^{\text{sim}}\right)^2}{\sum_{i=1}^{n}\left(x_i^{\text{obs}} - x_i^{\text{mean}}\right)^2} \right] \tag{1}$$

$$\text{PBIAS} = \left[ \frac{\sum_{i=1}^{n}\left(x_i^{\text{obs}} - x_i^{\text{sim}}\right)}{\sum_{i=1}^{n}\left(x_i^{\text{obs}}\right)} \times 100 \right] \tag{2}$$

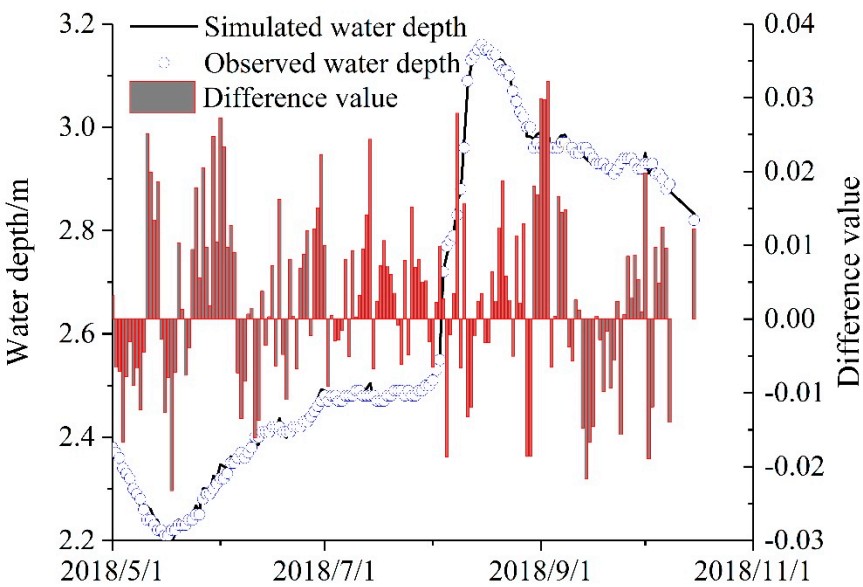

**Figure 4.** Comparison of simulated and observed water depth and their differences (simulation minus observations) of Chagan Lake from May–October 2018.

The measured and simulated values of salinity and temperature at six stations (S1–S6) covered three main regions of the lake, namely the western, middle, and eastern regions (Table 4 and Figure 5). The validation of salinity and temperature used the monthly measured data at three stations (A1–A3), covering the three main regions of the lake, is illustrated in Table 4 and Figure 5. The NSE values for salinity are all greater than 0.50, except at station S4 and S6 (Table 4). Overall, the simulated data matched well with field measurements at nine stations, thereby validating the integrated hydrodynamics-salinity model for simulating the salinity dynamics in Chagan Lake.

**Table 4.** Calibration and validation results of salinity and temperature in Chagan Lake.

| Sampling Stations | Salinity (psu) | | Temperature (°C) | |
|---|---|---|---|---|
| | NSE | PBIAS | NSE | PBIAS |
| Calibration (six observations for each station) | | | | |
| S1 | 0.62 | 5.88 | 0.75 | 1.65 |
| S2 | 0.51 | 4.30 | 0.94 | 1.65 |
| S3 | 0.68 | 10.54 | 0.94 | 3.88 |
| S4 | 0.41 | 7.15 | 0.99 | 0.68 |
| S5 | 0.69 | 9.98 | 0.94 | 2.97 |
| S6 | 0.38 | 5.01 | 0.74 | 2.47 |
| Validation (six observations for each station) | | | | |
| A1 | 0.81 | 2.76 | 0.99 | −0.56 |
| A2 | 0.77 | −0.54 | 0.99 | −0.76 |
| A3 | 0.76 | 1.71 | 0.99 | −0.67 |

**Figure 5.** Comparison of observed and simulated salinity and temperature in different stations from May to October 2018. Stations S1–S6 were used for calibration and stations A1–A3 were used for validation (Sim and Obs refers to simulated and observed data, respectively).

## 2.4. Irrigation Districts Development Scenarios

Multiple irrigation development scenarios were prepared based on two objectives. The first objective was to explore the salinity trend under different irrigation development areas. The second objective was to predict when the salinity of lake water wouldl reach the salinity threshold under different irrigation developments. Scenarios A–C were used to demonstrate the first objective. The existing irrigation district conditions were set at the baseline scenario (Scenario A). Current designed discharge volume from the Qianguo and Da'an irrigation districts in the baseline scenario were about 1.6

$\times$ 108 m$^3$/a and 0.91 $\times$ 108 m$^3$/a, respectively. Salinity in the discharges was obtained through monthly field experiments. For Scenario B, on the basis of the baseline scenario, 100% of irrigation discharge in the Qian'an irrigation district was discharged into Chagan Lake. Scenario C represented that 75% of irrigation discharge in the Qian'an irrigation district was discharged into Chagan Lake. The 100% and 75% irrigation discharge of Qian'an irrigation district were 0.60 $\times$ 10$^8$ m$^3$/a and 0.45 $\times$ 10$^8$ m$^3$/a, respectively. Scenarios A–C can intuitively represent the amount of irrigation discharge that Chagan Lake can currently tolerate. In addition, we can also find the sensitive areas in regard to the lake's response to the Qian'an irrigation discharge according to the change in salinity at different observation stations. Scenarios A–B were used to demonstrate the second objective based on the sensitive observation stations. The salinity threshold was set to 1 psu based on the regulation of the Fishery Water Quality Standard in China (GB11607-89). For the Scenario A analysis, a prediction was made regarding when the lake would reach threshold based on the baseline scenario. For Scenario B, the analysis predicted the time for the salinity to reach threshold.

## 3. Results

### 3.1. Long-Term Variation and Spatial-Temporal Distribution of Salinity in the Lake

As shown in Figure 6, the average mean value of water salinity in Chagan Lake shows increased from 2008 to 2018, having nearly doubled in salinty from 0.3 psu to 0.58 psu. The maximum salinity value (0.78 psu) occurred in 2012, and salinity continued to rise after temporarily falling in 2013 (0.5 psu). Seasonally, the salinity values in summer and autumn compared with those in spring at six stations (S1–S6; Figure 5). The salinity values were highest in June compared to other seasons at the A1 station (Figure 5). Salinity ranged from 0.17 psu to 0.89 psu, which had a peak value at station A1 on June 14 and a minimum value at station A3 on August 25 (Figure 5). As for the water temperature, in July it was greater than in other seasons at nine stations (Figure 5). The water temperature ranged from 5.71 °C to 30.80 °C, with the maximum and minimum value observed at station A1 on July 23 and October 31, respectively.

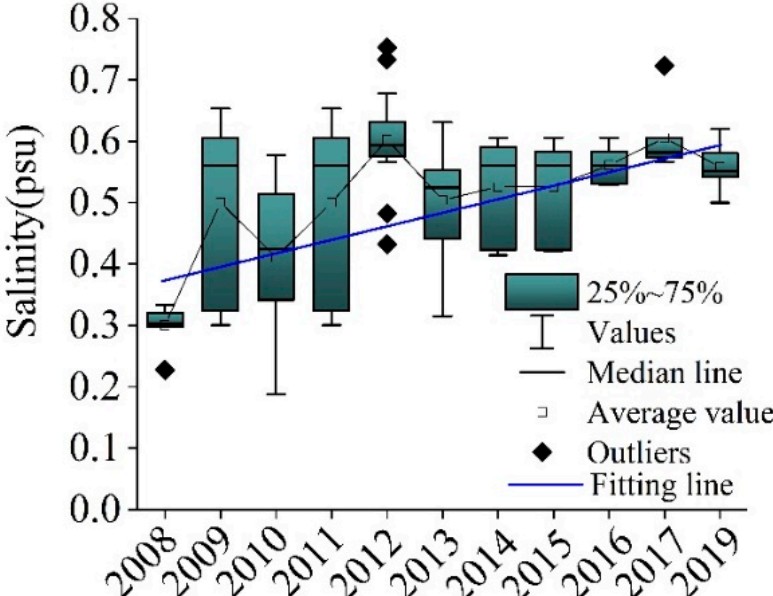

**Figure 6.** The tendency of salinity in Chagan Lake from 2008 to 2018. Each boxplot illustrates the median and inter-quartile range and the whiskers indicate minimum and maximum values. The blue solid line indicates the fitting line of salinity. The black points indicate the outliers.

The patterns of salinity in the study area shows uniform distribution in different months and regions (Figure 7). There was an increased tendency of salinity from the eastern to western regions in

the lake (Figure 7). The maximum value observed was in the area adjacent to the Jiangjia Weir, and the minimum salinity was adjacent to the Chuantou Weir. Water salinity in the western region of the lake was higher than those in the middle and eastern regions of the lake (Figure 5). The water temperatures varied insignificantly in different areas of the lake (Figure 5).

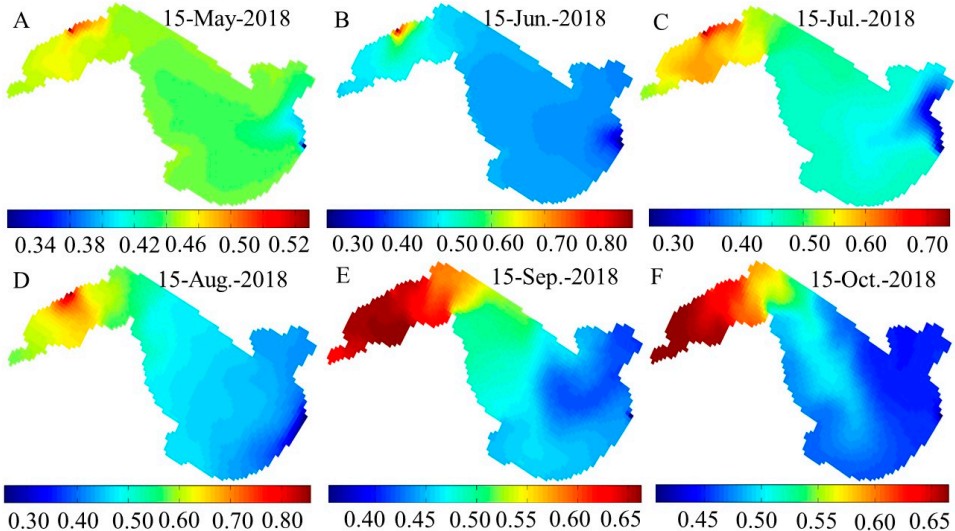

**Figure 7.** Spatial-temporal distribution of salinity from May to October in Chagan Lake. (**A**–**F**) indicate May to October.

### 3.2. Salt Accumulation in the Lake

Salt accumulation in the lake is also estimated based on the water quality data from May to October at the main inlet and outlet of the lake in 2018 (Figure 8). The salinity loading was found to attribute to input from the groundwater (37.25%), and drainages form the Da'an (41.67%) and the Qiangguo (21.08%) irrigation districts. Furthermore, the absorption, resuspension, and settlement of salinity (i.e., 71.86%) were based on the mass balance calculation. In terms of salinity, 83.85% of it flowed out the lake while 16.15% permeated into groundwater. The annual salt storage from May to October in the lake was 96.99 t.

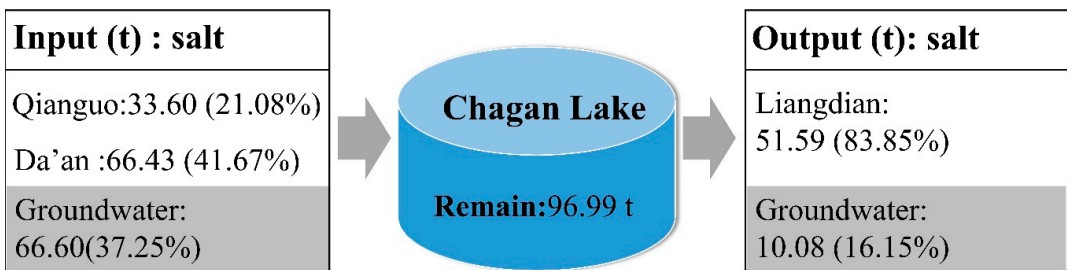

**Figure 8.** The overall mass balance for salt accumulation in Chagan Lake from May to October 2018.

### 3.3. Time to Reach Salinity Threshold for Freshwater Fish Farming

Simulated and predicted results are presented in Figures 9 and 10. A series of salinity without involvement of the Qian'an irrigation district was simulated within the threshold (1.0 psu; Figure 9A). Salinity at station A1 and S1 was significantly higher than those of other observations stations. In contrast, salinity at station A3 and A2 was much lower than those observed at other stations. The lake water salinity generally trended upwards from May to September, and then decreased from September to October at stations A1 and S1. Predicted salinity at station S5 exceeded the threshold over a short period from July to August with a 100% area of Qian'an irrigation district discharge

(Figure 9B). In this simulation scenario, the values at station S5 were much higher than those values at other stations. Predicted salinity at all stations was within the threshold when the 75% area of the Qian'an district drainage was included (Figure 9C). Salinity at station S5 was also higher than any other station except at station A1 from May to July. Figure 10b,c show that the salinity of station S5 was more sensitive to the lake salinity increasing with irrigation discharge of the Qian'an irrigation district.

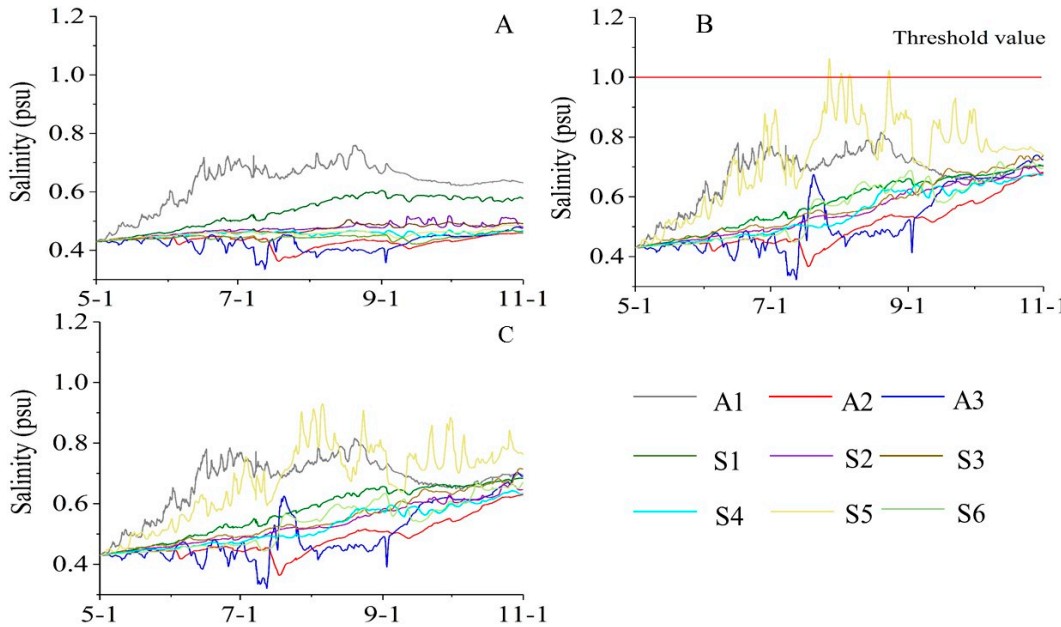

**Figure 9.** Salinity variation under nine stations of irrigation districts. (**A**)indicates existing irrigation districts; (**B**) indicates 100% area of Qian'an irrigation discharge; (**C**)indicates 75% area of Qian'an irrigation discharge.

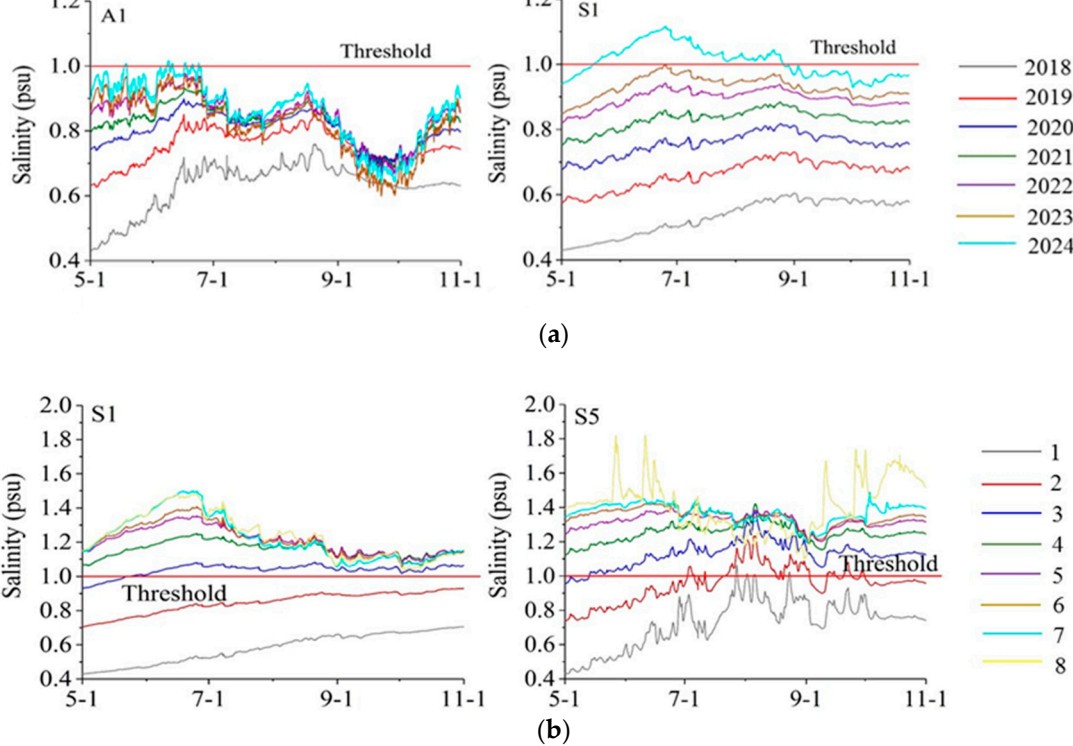

**Figure 10.** Time series prediction of salinity in the lake exceeding threshold under scenario (**a**) and scenario (**b**) at nine stations. A1, S1, and S5 indicate the observed stations.

The predictions of lake salinity increasing above the threshold in scenario (a) and scenario (b) at different station are presented in Figure 10, which shows that the increase in salinity was attributed to the irrigation discharge. Salinity of station A1 and S1 was higher than those of other stations (Figure 9A). Hence, we selected station A1 and S1 to predict the time when salinity reaches the threshold of 1 psu for freshwater fish farming under scenario (a). Lake salinity showed an increasing tendency after 2018 and the maximum values of station A1 and S1 will reach the threshold in 2024 (Figure 10a). Predicted salinity at station S1 and S5 was more sensitive than at other stations (Figure 9B). Hence, station S1 and S5 were chosen to predict time for exceeding the threshold (1 psu) for freshwater fish farming under scenario (b). Predicted salinity showed an increasing trend starting from the first year when Qian'an irrigation discharged into the lake. In addition, salinity at station 5 reached the threshold in the first year, and both station S1 and S5 exceeded the threshold in the third year (Figure 10b). The predicted peak value was 1.8 psu (Figure 10b) but the rate of increase was predicted to be moderate after the fifth year. These results indicate that the lake water salinity will increase to 1 psu in 2024 without involvement of the Qian'an district. The salinity will exceed 1 psu in three years if the irrigation water from the Qian'an irrigation district also drains into the lake.

## 4. Discussion

### 4.1. Spatial-Temporal Changes of Lake Water Salinity

Previous studies have shown that salinity decreases with depth and water temperature, that is, the salinity in summer is lower than in other seasons in the northern hemisphere [5,6,51,78]. However, this study showed that the salinity of Chagan Lake increases with temperature from spring to summer (Figure 5). Chagan Lake is located in arid and semi-arid areas and its evaporation is much higher than precipitation [62]. Its main supplement water is irrigation discharge, which is almost a semi-closed lake. The outlet of Chagan Lake is an overflow weir, and sufficient water volume and stable water level could be safeguarded by irrigation discharge in the summer. In addition, Chagan Lake is surrounded by saline-alkali land, and large amounts of salt and alkali are carried into Chagan Lake with irrigation discharge [58]. Therefore, there is higher salinity in Chagan Lake in the summer due to higher temperatures, greater evaporation, and more irrigation discharge. This is similar to the cases for lakes in arid and semi-arid regions [79–81]. Discharges in Qianguo and Da'an irrigation districts mainly occurred in May and September due to irrigation mode. [58]. The highest level of salinity in the lake entrance was in June (Figure 5) and the highest salinity of most lake regions was in summer (Figure 5), suggesting a possible lead-lag effect of irrigation discharges on salinity dynamics. In addition, the salinity of the saline-alkali land flowed into the irrigation channel and entered the lake due to the concentration precipitation in the summer [81,82].

Salt transport caused by irrigation discharge often aggravates water salinization in arid regions [83]. The new irrigation district such as the Da'an irrigation district should bring high salt concentrations from the saline-alkali land to the lake in a short time (Figures 7 and 8). Old irrigation districts such as the Qianguo irrigation district have low discharge salinity due to the long-term salt washing effect [19,84,85], but salinization is predicted to be exacerbated by the addition of the new irrigation district. Other environmental factors such as wind, may also have an impact on the salinization process. Wind typically accelerates the diffusion of ions in water by changing the hydrodynamics field [86,87]. The prevailing wind direction was southwestern, which may prevent salinity from spreading from the northwestern part of the lake to the eastern part of the lake (Figure 7). Nevertheless, this study demonstrates that evaporation and irrigation discharge are the main driving force for the seasonal variation of lake salinity in arid and semi-arid regions, while irrigation discharge is the main driving force for spatial variations of lake salinity.

### 4.2. The Effect of Salinization on Lake Ecosystem Health

Lake salinization is particularly prone to damaging freshwater ecosystems, particularly destroying the biological communities of microbes [88–90], plankton [29,30,82,91], macrophytes [92], fish [93], mammals [88], and water birds [93]. According to Hammer [56], saline lakes have a salinity level equal to or in excess of 3.00 psu, freshwater lakes have dissolved salt concentrations of less than 0.5 psu, and sub-saline lakes have a salinity level between 0.5 and 3.0 psu. Based on this classification, Chagan Lake may eventually evolve from a freshwater lake to sub-saline lake, similar to the process of salinization of the Boston Lake [28,51]. However, the depth of Chagan Lake is less than Boston Lake [51,61], hence the ecosystem of Chagan Lake is more vulnerable to water salinization. The fish community structure and production are limited by the increased salinity of the lake because higher salinity can inhibit fish embryos from hatching and can reduce plankton production [90,93]. Our results have shown that salinity is relatively lower in the eastern region of Chagan Lake than in the western region (Figures 5 and 7), indicating that fishery development may be restricted in the western region. In addition, excessive irrigation discharge would accelerate lake salinity and thus reach the threshold for freshwater fish farming (Figures 9 and 10). This finding (i.e., the effects of irrigation discharge on the spatial-temporal dynamic of lake salinity) is critical to policy making for sustainable water management in Chagan Lake.

### 4.3. Strategy for Sustainable Lake Water Management to Control Salinization

A number of short-term and long-term salinity management strategies have been developed around the world [94,95], promoting the balance of regional ecology and the economy. Previous studies have shown that conflicts of interest between ecological health of the lake and its surrounding irrigation areas of development exist worldwide [6,61,62,96]. Accelerating the lake water circulation and decreasing the salinity loading from the input water are two effective methods to maintain the salinity below the threshold [97]. Economic benefit also needs to be taken into account as well as the effect on ecosystem health [98]. Thus, it is extremely important to carry out hydrological and economic analysis in areas with respect to both water shortage and salinization [99]. Reducing the salinity of a lake is an urgent problem for the lakes surrounded by irrigation areas. Reducing the salinization of lakes by increasing water volume has been applied to lakes where the volume of water varies greatly [100,101]. However, the water level of Chagan Lake is stable [58], which means Chagan Lake salinity in water sources has an even more critical impact. Water diversion and accelerated lake water renewal can effectively control the concentration of lake pollutants, thereby maintaining the ecological health of the lakes [102]. Hence, it may be effective to reduce lake salinity by changing the hydrodynamics force and decreasing the stagnant time of lake water. In addition, desalination for irrigation discharge using wetlands has the potential for lake salinification control [103]. A study by Yang [39] showed that the amount of salt removed by harvesting reeds and cattails accounted for 10–26% of the discharged salt concentration. Thus, salinity of the western region of the lake was lower than that of the eastern region (Figure 7) which attributed to the effect of the wetland.

In summary, developing sustainable management for irrigation discharge is crucial to prevent inland freshwater lakes from further salinization, especially those lakes surrounded by irrigation areas. The following specific suggestions of mitigation measures are meant to alleviate salinization of Lake Chagan: (1) implement multi-source diversion measures from the Songhua River for diluting lake salinity; (2) enhance the salinity purification function of the front wetland by expanding the planting area of reed. However, caution should be taken so that the simulated and predicted salinity values in this study are not obtained with significant amounts of hourly water quality data and high-density monitoring station networks, as the lack of these can cause uncertainties in our modeling results.

## 5. Conclusions

This case study simulated long-term salinity variations using field monitoring datasets and assessed the effects of irrigation discharge on spatial-temporal variations of salinity in Chagan Lake using an integrated hydrodynamics-salinity model. The results support the hypothesis that irrigation discharge causes lake salinization, especially in the western region of Chagan Lake adjacent to the new irrigation district. Our model simulations show that Chagan Lake will continue to accumulate salt, and that the annual average salinity of the lake will reach the threshold of 1 psu which threatens the current freshwater fish farming. Based on these findings, we stress the importance of considering water diversion and wetland purification in the arid and semi-arid area to protect degradation of lake ecosystem functions, especially to prevent lake salinization and to sustain freshwater fish farming.

**Data Availability Statement:** Some or all data, models, or code generated or used during the study are available from the corresponding author by request.

**Author Contributions:** X.L. compiled the literature and wrote the whole literature; G.Z. and J.Z. designed the structure of the literature; Y.J.X., G.S. and Y.W. (Yanfeng Wu) polished the paper; Y.W. (Yao Wu), X.L., Y.C. and H.M. collected the water samples. All authors have read and agreed to the published version of the manuscript.

**Funding:** This research was supported by the National Key R&D Program of China (2017YFC0406003); the National Natural Science Foundation of China (41877160); the National Key R&D Program of Jilin Province.

**Conflicts of Interest:** The authors declare no conflict of interest.

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
