# Peer review of "Effects of Irrigation Discharge on Salinity of a Large Freshwater Lake: A Case Study in Chagan Lake, Northeast China"

_water, doi:10.3390/w12082112_

Round 1
Reviewer 1 Report
The work presents an analysis quantifying the effects of irrigation withdrawn in salinity regime for Changan lake (China) . The manuscript is well written and clearly defined the key idea and the results. I ,therefore, propose it for publishing after minor revisions. I want the authors to spend more time and effort in describing thorough the methodology followed. A generic description is provided in chapter 2.2 and 2.3.1 and I wish to see more details therein.
Author Response
[Water] Manuscript ID: water-831030
Effects of irrigation discharge on salinity of a large freshwater lake: A case study in Chagan Lake, Northeast China
Reviewers' comments:
Reviewer1#
The work presents an analysis quantifying the effects of irrigation withdrawn in salinity regime for Changan lake (China). The manuscript is well written and clearly defined the key idea and the results. I, therefore, propose it for publishing after minor revisions. I want the authors to spend more time and effort in describing thorough the methodology followed. A generic description is provided in chapter 2.2 and 2.3.1 and I wish to see more details therein.
[Authors’ response] Thank you very much for your support and for taking time to review our manuscript. Your comments and suggestions are helpful for us to improve our manuscript, all of which have been taken into account in our revised manuscript. We have refined the description of chapter 2.2 and 2.3.1. We divided chapter 2.2 into 2.2.1 and 2.2.2.
Specifically as following:
2.2 Data collection and model inputs
2.2.1 Sample collections (lines 125-142)
2.2.2 Model inputs (lines 143-154)
The meteorological datasets included (hourly wind speed, wind direction, precipitation, evaporation, solar radiation, cloud cover, relative humidity and air temperature, etc.) in 2018 were derived from Qianguo meteorological station (http://data.cma.cn/) (Table 2), which have been used as driven forcing inputs into the model. Parameters such roughness, Dalton, viscosity and Stanton number used in the hydrodynamic model were obtained from literatures [67-68] (Table 2). These datasets of salinity, temperature and water levels between May and October in 2018 were used for the model calibration and validation, respectively. In addition, geometric data including bathymetry and land boundaries were obtained based on the google map. The tools of RGFGRID and QUICKIN provided by Delft3D software suite have been used to create grids and incorporate bathymetry data (Fig. 2). A grid of finite difference quadrangular elements was created for the whole Chagan Lake with 4928 cells as showed in Fig. 2.
In addition, we have refined the description of chapter 2.3.1 in lines 173-183.
2.3.1 Modeling setup
Initial condition and stability: The observations of water level, salinity and temperature in May of 2018 were first used as initial conditions. In order to stabilize the model, the spin-up time of the model was 4 months (from January to April) with the time step 1 minuite. Then, the initial conditions were reset with a restart file from this warm-up run. The model stability was also checked with the continuity check suggested by the delft3D user manual with a constant value of 1.0 over the simulation period.
Model assumption: Given the lake is very shallow with an average depth less than 2.5 meter, it was assumed the lake being well-mixed with no vertical stratification, and a 2D- hydrodynamic model was applied. The simulation period was 6-moonth between May and October, whereas the frozen-period between November and April was assume to have no influence on the results (no inflow and outflow into or out of the lake during this period).
More description is in line 208-212.
2.3.3 Calibration and validation
Model calibration and validation were conducted by comparing simulated results with the observations [53, 74] from May to October in 2018. Given the higher uncertainties in the limited dataset, the calibration was mainly through adjusting inflow and outflow rates. The parameters such as Roughness, Dalton, Viscosity and Stanton number obtained from literatures were slightly adjusted as presented in Table 2.
Thank you so much, again, for taking time to thoroughly review our manuscript. We greatly appreciate you for the helpful comments and suggestions.

Reviewer 2 Report
The manuscript “Effects of irrigation discharge on salinity of a large freshwater lake: A case study in Chagan Lake, Northeast China” by Liu and co-authors presents valuable and timely information on long-term salinity variations (in time and space) in Chagan Lake (China) promoted by irrigation discharges. A model was used to explain the results and gave rise to important predictions. Although the results are interesting and important, the paper needs a throughout revision for clarity. Materials and Methods, and also Results section, need to be reduced and focused. The number of figures must to be reduced. The results would profit from a thoroughly, more complete, discussion. Mitigation measures seem underexplored. An English revision by a native speaker is mandatory. Some terms also need to be clarified (please see specific comments). I’ve included some comments /suggestions below that, I hope, may be of use.
Specific comments
Title: do you mean “a case study in Changan Lake” or “the case study of Changan Lake” (?)
Abstract
Line 19. effects or affects?
Line 29. “up to a mean of 0.62 psu”. Mean of what? Summer and autumn data?
Line 32-34. The threshold value should be indicated.
Introduction.
Lines 42-43. Too general. I suggest that the authors indicate, briefly, the effects of salinity on lakes. You do have several lines in this and the next paragraph on salinization origin.
Line 47. Remove extra comma.
Line 48. Remove “into lakes”. Redundant.
Lines 49-50. This is an important issue that could be considered in the discussion (or a little developed in the introduction) that would underline the global importance and interest on your study. Do you mean “district” or, less specific, “area”? In my opinion, district is a specific area (administrative, for instance) in a country. The difference, and though specific use of the term, should be clarified throughout the text.
Lines 53-56. The sentence should be rephrased and eventually shortened.
Lines 59-61. So, is the reduction of O2 a positive response?
Lines 62-63. Not conductivity?
Lines 64-65. I believe that, at this point, you are mentioning lakes and not a specific lake.
Line 67. “of irrigation and discharge areas”?
Line 83. Top ten. In area? Clarify.
Line 95. “timing for reaching the salinity threshold for freshwater aquaculture”. Which is….?
Lines 95-95. Does the lake provide freshwater to the population? If so, salinity values should also consider that. That should me mentioned in the discussion.
Materials and Methods & Results
Both sections should be condensed and focused. Number of figures should be reduced and legends reviewed.
Some inaccuracies need to be corrected. For example, the verbs tense (throughout the text)
Line 111. are presented
Line 113. “Chagan Lake would also receive irrigation district from the Qian’an irrigation district in the future.”???
Table 1. Legends: What do you mean by “basic”? Align parameters with means and range. Note: “Indicate significance testing of means”? please rephrase. the word derived in not appropriate
Line 266-267. “The loading of salinity from groundwater FROM Da’an irrigation districts AND Qianguo districts were…”
Line 273. “When the salinity achieved…” does not seem an appropriate title. Rephrase
Line 274. The results “are” presented, not “were”
Discussion
Excessive number of figures in the manuscript and no figures should appear in the discussion (e.g., Fig 12).
I missed some considerations on:
Does the frozen period have an effect on salinization levels/patterns?
Can salinity also increase due to aquaculture activities?
Effects on the ecosystem health itself and population health (use of water to drink)
Specific suggestion of mitigation measures.
The manuscript considers the importance of salinity to fisheries /aquaculture which, in my opinion, are not synonyms.
What about effects of salinity on the ecosystem functioning? It may deserve some development.
Lines 311-312. “Previous studies have shown that salinity decreases with increased water level (OK!) and water temperature (?), that is, the salinity in summer (?) was lower than in other seasons in the northern hemisphere. This sentence is general and refers to previous studies. If I understand correctly, in northern hemisphere, in summer we have higher temperatures and higher water evaporation. Shouldn’t salinity increase? Please clarify
Lines 319-320. “The time… mode”. The sentence needs to be rephrased; not clear what you mean
Lines 333-334. “The lower…washing”. The sentence needs to be rephrased; not clear what you mean
Line 344. This section should be developed. Several issues need to be developed. Does the frozen period have an effect on salinization levels/patterns?
Can salinity also increase due to aquaculture activities?
Line 362. “the ecological health of lakes (in general, right?)”. In any case if you refer irrigation districts it means that lakes receive irrigation discharge. The last part of the sentence is redundant.
Line 366. also needs?
Line 368. Was or is ? The authors should pay attention in the use of past or present.
Line 381-384. “by the saline irrigation districts”. Why? Include a brief explanation and explain (briefly) differences in relation to other contexts.
Author Response
[Water] Manuscript ID: water-831030
Effects of irrigation discharge on salinity of a large freshwater lake: A case study in Chagan Lake, Northeast China
Reviewers' comments:
Reviewer2#
The manuscript “Effects of irrigation discharge on salinity of a large freshwater lake: A case study in Chagan Lake, Northeast China” by Liu and co-authors presents valuable and timely information on long-term salinity variations (in time and space) in Chagan Lake (China) promoted by irrigation discharges. A model was used to explain the results and gave rise to important predictions. Although the results are interesting and important, the paper needs a throughout revision for clarity. Materials and Methods, and also Results section, need to be reduced and focused. The number of figures must to be reduced. The results would profit from a thoroughly, more complete, discussion. Mitigation measures seem underexplored. An English revision by a native speaker is mandatory. Some terms also need to be clarified (please see specific comments). I’ve included some comments /suggestions below that, I hope, may be of use.
[Authors’ response] Thank you very much for your support and for taking time to review our manuscript. We have made a deal of effort to check on the English grammar and usage, and have tried our best to make our paper more readable. The final version of the manuscript has been proofread by a proficient English writer. Your comments and suggestions are helpful for us to improve our manuscript, all of which have been taken into account in our revised manuscript.
Specific comments
Title: do you mean “a case study in Changan Lake” or “the case study of Changan Lake” (?)
[Authors’ response] Thanks for catching this. We mean “a case study in Changan Lake” in the title.
Abstract
Line 19. effects or affects?
[Authors’ response] Sorry for the mistake. It should be “affects”.
Line 29. “up to a mean of 0.62 psu”. Mean of what? Summer and autumn data?
[Authors’ response] Sorry for the confusion. We have rewritten the sentence as following (lines 30-31):
and (ii) between May to October the salinity was up to 0.62 psu in the western region of the lake.
Line 32-34. The threshold value should be indicated.
[Authors’ response] We fully agree. The threshold (1 psu) has been indicated in line 25.
Introduction.
Lines 42-43. Too general. I suggest that the authors indicate, briefly, the effects of salinity on lakes. You do have several lines in this and the next paragraph on salinization origin.
[Authors’ response] Thanks for your good suggestion. We indicate, briefly, the effects of salinity on lakes in lines 42-43. Specifically as following:
The salinization of freshwater has received increasing attention because it can damage the ecosystems services, mainly prone to damaging biological communities [1-5].
In addition, we have described the lake salinization origin in this paragraph, such as climatic processes, agricultural production, industrial sewage etc. lines 43-48. Specifically as following:
Although lake water salinities may vary cyclically with climatic processes [6], significant salinization in the past decades was largely due to anthropogenic activities, such as agricultural production [7-10], industrial sewage [11], human-accelerated weathering [12-13], and land clearing [14-15].
Line 47. Remove extra comma.
[Authors’ response] Done (line 46)
Line 48. Remove “into lakes”. Redundant.
[Authors’ response] Done (line 48)
Lines 49-50. This is an important issue that could be considered in the discussion (or a little developed in the introduction) that would underline the global importance and interest on your study. Do you mean “district” or, less specific, “area”? In my opinion, district is a specific area (administrative, for instance) in a country. The difference, and though specific use of the term, should be clarified throughout the text.
[Authors’ response] We fully agree. The effects of agricultural activities on salinization is an important issue. We have developed in the introduction (lines 52-58). Specifically as following:
A number of studies have shown that agricultural activities, which using pesticides and fertilizers, groundwater extraction for irrigation, unreasonable irrigation and drainage methods, land-use change, have accelerated accumulation of salt, causing soil salinization [18-20] and groundwater salinization [21-23], which are believed to eventually bring about reductions in cropping productivities [24-25]. These studies focused on the effects of agricultural activities on soil salinization and water body salinization of irrigation area. It is also know that the irrigation discharge would increase the lake salinity.
In addition, we also considered irrigation discharge would change the spatio-temporal pattern of lake salinity in the discussion (lines 336-345). Specifically as following:
In addition, Chagan Lake is surrounded by saline-alkali land, and large amounts of salt and alkali are carried into Chagan Lake with irrigation discharge [58]. Therefore, there was higher salinity of Chagan Lake in summer due to the higher temperature, the greater evaporation, and the more irrigation discharge, which was similar to the cases for lakes in arid and semi-arid regions [79-81]. Discharges in Qianguo and Da’an irrigation districts mainly occurred in May and September due to irrigation mode. [58]. While, the higher salinity of the entrance was in June (Fig. 5) and the higher salinity of most lake regions happened in summer (Fig. 5), suggesting a possible lead-lag effect of irrigation discharges on the salinity dynamics. In addition, the salinity of the saline-alkali land flowed into the irrigation channel and entered the lake due to the concentration precipitation in summer [81-82].
We are sorry for the confusion. We have clarified throughout the full text. Chagan Lake is affected by several irrigation district including Qianguo irrigation district, Qian’an irrigation district and Da’an irrigation district. The “district” refer to special irrigation district such as Hetao Irrigation District (Yu et al., 2010); while the “area” refer to general irrigation area. In addition, the “discharge area” refer to the area receiving irrigation discharge (line 59).
Yu, R.; Liu, T.; Xu, Y.; Zhu, C.; Zhang, Q.; Qu, Z.; Li, C. Analysis of salinization dynamics by remote sensing in Hetao Irrigation District of North China. Agr. Water Manage. 2010, 97(12), 1952-1960.
Lines 53-56. The sentence should be rephrased and eventually shortened.
[Authors’ response] Done in lines 58-60. Specifically as following:
However, these studies rarely addressed the effect of irrigation discharge on discharge areas (the area receiving irrigation discharge), especially long-term simulation and prediction.
Lines 59-61. So, is the reduction of O2 a positive response?
[Authors’ response] Sorry for the confusion. This sentence indicates that the water diversion project has caused the density stratification of deep-water lakes, which has led to the decreasing of dissolved oxygen in the lake. In the management of lake salinization using the water diversion project, other water quality indicators should be comprehensively considered. In addition, it is better to use the water diversion project to control shallow lake salinization.
Lines 62-63. Not conductivity?
[Authors’ response] Sorry for the negligence. We have added the conductivity in lines 65-67. Specifically as following:
Most previous studies on lake salinization used Cl-, total dissolved solids (TDS) or electrical conductivity (EC), rather than salinity, to measure how irrigation discharge affect freshwater lake salinity [33-35].
Lines 64-65. I believe that, at this point, you are mentioning lakes and not a specific lake.
[Authors’ response] We fully agree. We have corrected in line 67-68. Specifically as following:
Whilst, some researchers demonstrated that the Cl- and TDS cannot replicate inherent variation of salinity in freshwater lake, especially in the inland waters [36-37].
Line 67. “of irrigation and discharge areas”?
[Authors’ response] Sorry for the mistake. We have corrected in lines 68-70. Specifically as following:
Therefore, further investigation is needed to see how and to what extent freshwater lake salinity have changed and will evolve with consideration of irrigation areas.
Line 83. Top ten. In area? Clarify.
[Authors’ response] Sorry for the confusion. We have clarified it in lines 85-87. Specifically as following:
Located in Northeast China, Chagan Lake is among the ten largest freshwater lakes in China in terms of area, and it is an important base of freshwater fishery [58].
Line 95. “timing for reaching the salinity threshold for freshwater aquaculture”. Which is….?
[Authors’ response] Sorry for the confusion. This means a time when the salinity reach a threshold (1psu) for freshwater fish farming (lines 96-97).
Lines 95-95. Does the lake provide freshwater to the population? If so, salinity values should also consider that. That should me mentioned in the discussion.
[Authors’ response] Thank you for the good suggestion. However, Chagan Lake does not provide freshwater to the population.
Materials and Methods & Results
Both sections should be condensed and focused. Number of figures should be reduced and legends reviewed.
[Authors’ response] Your point is well taken. We have merged Figure 5 and Figure 6, and deleted Figure 12.
Figure 5. Comparison of observed and simulated salinity and temperature in different stations from May to October in 2018. Stations S1-S6 for calibration and stations A1-A3 for validation (Sim and Obs refer to simulated and observed data, respectively).
Some inaccuracies need to be corrected. For example, the verbs tense (throughout the text)
[Authors’ response] Sorry for the mistakes. We have revised the verbs tense in the full text. Specifically as following table:
|
Original |
Revised |
Line |
Original |
Revised |
Line |
|
were |
are |
114 |
showed |
shows |
270 |
|
were |
are |
294 |
was |
is |
284 |
|
would |
will |
409 |
were |
are |
311 |
|
was |
is |
395 |
received |
receive |
113 |
|
receive |
received |
81 |
showed |
shows |
261 |
|
were |
are |
71 |
simulate |
simulated |
93 |
|
variation |
variations |
94 |
to predict |
predict |
96 |
|
is causing |
has caused |
107 |
was |
is |
107 |
|
received |
receives |
110 |
increased |
increases |
112 |
|
have become |
are becoming |
50 |
is |
was |
181 |
|
showed |
show |
305 |
reached |
reaches |
314 |
|
aggravated |
aggravats |
347 |
prevented |
prevent |
354 |
|
were |
are |
356 |
was |
is |
357 |
|
evolved |
evolve |
365 |
was |
is |
367 |
|
were |
are |
368 |
showed |
have shown |
370 |
|
was |
is |
370 |
was |
is |
396 |
Line 111. are presented
[Authors’ response] Done
Line 113. “Chagan Lake would also receive irrigation district from the Qian’an irrigation district in the future.”???
[Authors’ response] Sorry for the confusion. Chagan Lake now receives irrigation discharge from Qianguo and Da’an irrigation districts, but Chagan Lake is to receive irrigation district from the Qian’an irrigation district in the future, according to the development plan of Songyuan irrigation district (lines 112-113).
Table 1. Legends: What do you mean by “basic”? Align parameters with means and range. Note: “Indicate significance testing of means”? please rephrase. the word derived in not appropriate
[Authors’ response] Sorry for the confusion caused by “basic”. We have deleted it in line 117. We mean the physical parameters and coefficients used in the integrated hydrodynamics-salinity model in Table 1. In addition, number a and b represents significance levels of these parameters passing the 0.0001 and 0.001 significance test, respectively. (lines 118-119)
Line 266-267. “The loading of salinity from groundwater FROM Da’an irrigation districts AND Qianguo districts were…”
[Authors’ response] We have corrected it in lines 285-287. Specifically as following:
The salinity loading was found to attribute to input from the groundwater (37.25%), and drainages form the Da’an (41.67%) and the Qiangguo (21.08%) irrigation districts.
Line 273. “When the salinity achieved…” does not seem an appropriate title. Rephrase
[Authors’ response] Thank you for the good suggestion. We have corrected the title in line 293. Specifically as following:
Time to reach salinity threshold for freshwater fish farming.
Line 274. The results “are” presented, not “were”
[Authors’ response] Done
Discussion
Excessive number of figures in the manuscript and no figures should appear in the discussion (e.g., Fig 12).
[Authors’ response] We fully agree. We have deleted it.
I missed some considerations on:
Does the frozen period have an effect on salinization levels/patterns?
[Authors’ response] Chagan Lake has no replenishment sources other than precipitation during the frozen period(Zhang et al., 2017), and groundwater replenishment is negligible. Thus, there is no extraneous salt flowing into the lake during frozen period. In addition, there is less evaporation on the ice surfaces during the frozen period (Haynes et al., 1992; Pratte et al., 2006), resulting in the slightly salinity enriched of the lake. However, the salinity change is relatively small in the overall lake. Therefore, we mainly study the effects of irrigation discharge on the lake salinity during non-frozen period. We have added the non-frozen in chapter 2.2.1.(line 127)
Zhang, L.; Hipsey, M. R.; Zhang, G. X.; Busch, B.; Li, H. Y. Simulation of multiple water source ecological replenishment for Chagan Lake based on coupled hydrodynamics and water quality models. Water Sci. Technol. 2017, 17(6), 1774-1784.
Haynes, D. R.; Tro, N. J.; George, S. M. Condensation and evaporation of water on ice surfaces. J. Phys. Chem. 1992, 96(21), 8502-8509.
Pratte, P.; van den Bergh, H.; Rossi, M. J. The kinetics of H2O vapor condensation and evaporation on different types of ice in the range 130− 210 K. J. Phys. Chem. A, 2006, 110(9), 3042-3058.
Can salinity also increase due to aquaculture activities?
[Authors’ response] The freshwater fish farming is one of the main ecosystem service of Chagan Lake. The lake salinization has a greater impact on freshwater fish farming. Therefore, we mainly discuss the importance of salinity control within the threshold for freshwater fish farming. There are less literatures on the effect of freshwater fish farming on salinity increasing, but there are many literatures that lake salinization has a negative effect on the growth and reproduction of fish (Ringø and Strøm, 1994; Boeuf and Payan, 2001; Zhao et al., 2020; Mukherjee et al., 2020).
Ringø, E.; Strøm, E. Microflora of Arctic charr, Salvelinus alpinus (L.): gastrointestinal microflora of free‐living fish and effect of diet and salinity on intestinal microflora. Aquac. Res. 1994, 25(6), 623-629.
Boeuf, G.; Payan, P. How should salinity influence fish growth?. Comparative Biochemistry and Physiology Part C: Toxicol. Pharmacol. 2001, 130(4), 411-423.
Zhao, R.; Symonds, J. E.; Walker, S. P.; Steiner, K.; Carter, C. G.; Bowman, J. P.; Nowak, B. F. Salinity and fish age affect the gut microbiota of farmed Chinook salmon (Oncorhynchus tshawytscha). Aquaculture 2020, 735539.
Mukherjee, P.; Gorain, P. C.; Paul, I.; Bose, R.; Bhadoria, P. B. S.; Pal, R. Investigation on the effects of nitrate and salinity stress on the antioxidant properties of green algae with special reference to the use of processed biomass as potent fish feed ingredient. Aquacult. Int. 2020, 28(1), 211-234.
Effects on the ecosystem health itself and population health (use of water to drink)
[Authors’ response] Chagan Lake is not a source of drinking water for the surrounding residents. Its main ecological services are tourism, freshwater fish farming and providing habitat. Therefore, the salinization of Chagan Lake has little effect on human drinking water. However, lake salinization would damage other ecosystem services, such as destroying biological communities.
Specific suggestion of mitigation measures.
[Authors’ response] More in line with the local governance measures, Chagan Lake mainly based on the existing water diversion project and wetland dilution and purification, specifically in lines 397-400:
Specific suggestion of mitigation measures are produced to alleviate salinization of Chagan:
1) Implement multi-source diversion measures from the Songhua River for diluting the lake salinity. 2) To enhance the salinity purification function of the front wetland by expanding the planting area of reed.
The manuscript considers the importance of salinity to fisheries /aquaculture which, in my opinion, are not synonyms.
[Authors’ response] Sorry for the confusion. The main ecosystem service of Chagan Lake is freshwater fish farming. Thus, we have changed the full text to indicate the importance of salinity changes for freshwater fish farming in line 97, 33, 314, 317, 372, 410, 413.
What about effects of salinity on the ecosystem functioning? It may deserve some development.
[Authors’ response] Thank you for the good suggestion. We have development discussed the effects of salinity on the ecosystem function in lines 359-361. Specifically as following:
Lake salinization would be particularly prone to damaging the ecosystem services of freshwater ecosystems, particularly destroying the biological communities of microbes [88-90], plankton [29-30, 91-92], macrophytes [93], fish [94], mammals [88] and water birds [94] etc.
Hart, B. T.; Bailey, P.; Edwards, R.; Hortle, K.; James, K.; McMahon, A.; Swadling, K. Effects of salinity on river, stream and wetland ecosystems in Victoria, Australia. Water Res. 1990, 24(9), 1103-1117.
References added
Jiang, H.; Dong, H.; Yu, B.; Liu, X.; Li, Y.; Ji, S.; Zhang, C. L. Microbial response to salinity change in Lake Chaka, a hypersaline lake on Tibetan plateau. Environ. microbial. 2007, 9(10), 2603-2621.
Jeppesen, E.; Brucet, S.; Naselli-Flores, L.; Papastergiadou, E.; Stefanidis, K.; Noges, T.; Bucak, T. Ecological impacts of global warming and water abstraction on lakes and reservoirs due to changes in water level and related changes in salinity. Hydrobiologia, 2015, 750(1), 201-227.
Schallenberg, M.; Hall, C. J.; Burns, C. W. Consequences of climate-induced salinity increases on zooplankton abundance and diversity in coastal lakes. Mar Ecol. Prog Ser. 2003, 251, 181-189.
Wollheim, W. M.; Lovvorn, J. R. Effects of macrophyte growth forms on invertebrate communities in saline lakes of the Wyoming High Plains. Hydrobiologia 1996, 323(2), 83-96.
Blinn, D., Halse, S., Pinder, A., Shiel, R. Diatom and micro-invertebrate communities and environmental determinants in the western Australian wheatbelt: a response to salinization. Hydrobiologia 2004, 528(1-3), 229-248.
Hintz, W. D.; Relyea, R. A. A salty landscape of fear: responses of fish and zooplankton to freshwater salinization and predatory stress. Oecologia 2017, 185(1), 147-156.
Senner, N. R.; Moore, J. N.; Seager, S. T.; Dougill, S.; Kreuz, K.; Senner, S. E. A salt lake under stress: Relationships among birds, water levels, and invertebrates at a Great Basin saline lake. Biol. Conserv. 2018, 220, 320-329.
Lines 311-312. “Previous studies have shown that salinity decreases with increased water level (OK!) and water temperature (?), that is, the salinity in summer (?) was lower than in other seasons in the northern hemisphere. This sentence is general and refers to previous studies. If I understand correctly, in northern hemisphere, in summer we have higher temperatures and higher water evaporation. Shouldn’t salinity increase? Please clarify
[Authors’ response] Sorry for the confusion. This sentence mainly describes that the lake’s water level would raise with higher precipitation and surface runoff in summer, in the northern hemisphere. In addition, the precipitation is much greater than the evaporation in most areas of the northern hemisphere (Deng et al., 2018). The effect of precipitation on salinity dilution is greater than that of evaporation on salinity concentration in these areas (Yihdego and Webb, 2012). Therefore, open lakes have lower salinity in humid areas, in summer. However, Chagan Lake is located in arid and semi-arid areas, which its evaporation is much larger than precipitation (Liu et al., 2020). Its main supply water is irrigation discharge, which is almost a semi-closed lake. The outlet of Chagan Lake is an overflow weir, and sufficient water volume and stable water level could be safeguarded by irrigation discharge in summer. In addition, Chagan Lake is surrounded by saline-alkali land, and large amounts of salt and alkali are carried into Chagan Lake in the irrigation discharge. Therefore, Chagan Lake has higher salinity in summer due to higher temperature, greater evaporation, and more irrigation discharge.
Specifically as following lines 333-345:
Chagan Lake is located in arid and semi-arid areas; its evaporation much higher than precipitation [62]. Its main supplement water is irrigation discharge, which is almost a semi-closed lake. The outlet of Chagan Lake is an overflow weir, and sufficient water volume and stable water level could be safeguarded by irrigation discharge in summer. In addition, Chagan Lake is surrounded by saline-alkali land, and large amounts of salt and alkali are carried into Chagan Lake with irrigation discharge [58]. Therefore, there was higher salinity of Chagan Lake in summer due to the higher temperature, the greater evaporation, and the more irrigation discharge, which was similar to the cases for lakes in arid and semi-arid regions [79-81]. Discharges in Qianguo and Da’an irrigation districts mainly occurred in May and September due to irrigation mode. [58]. While, the higher salinity of the entrance was in June (Fig. 5) and the higher salinity of most lake regions happened in summer (Fig. 5), suggesting a possible lead-lag effect of irrigation discharges on the salinity dynamics. In addition, the salinity of the saline-alkali land flowed into the irrigation channel and entered the lake due to the concentration precipitation in summer [81-82].
Deng, K.; Yang, S.; Ting, M.; Tan, Y.; He, S. Global monsoon precipitation: Trends, leading modes, and associated drought and heat wave in the Northern Hemisphere. J. Climate, 2018, 31(17), 6947-6966.
Yihdego, Y.; Webb, J. Modeling of seasonal and long-term trends in lake salinity in southwestern Victoria, Australia. J. Environ. Manage. 2012, 112, 149-159.
Liu, X.; Zhang, G.; Xu, Y. J.; Wu, Y.; Liu Y.; Zhang H. B. Assessment of water quality of best water management practices in lake adjacent to the high-latitude agricultural areas, China. Environ. Sci. Pollut. R. 2020, 27(3): 3338-3349.
Lines 319-320. “The time… mode”. The sentence needs to be rephrased; not clear what you mean
[Authors’ response] Sorry for the confusion. We have corrected in lines 340-341. Specifically as following:
Discharges in Qianguo and Da’an irrigation districts mainly occurred in May and September due to irrigation mode. [58].
Lines 333-334. “The lower…washing”. The sentence needs to be rephrased; not clear what you mean
[Authors’ response] Sorry for the confusion. We have corrected in lines 348-350. Specifically as following:
Old irrigation district such as Qianguo irrigation district has low salinity of the discharge due to the long-term salt washing effect [19, 84, 85], but salinization is predicted to exacerbate by the addition of the new irrigation district.
Line 344. This section should be developed. Several issues need to be developed. Does the frozen period have an effect on salinization levels/patterns?
[Authors’ response] Thank you for the good suggestion. Chagan Lake had no supplemental water sources other than precipitation during the frozen period (Zhang et al., 2017), and groundwater was negligible, so there was no source of extraneous salt. In addition, the amount of evaporation on the ice surfaces during the frozen period was small (Haynes et al., 1992; Pratte et al., 2006), resulting in a slightly enriched lake water salinity. However, the overall salinity change of the lake water is relatively small, so we mainly study the salinity change of the non-ice-bound period during the farmland withdrawal period.
Zhang, L.; Hipsey, M. R.; Zhang, G. X.; Busch, B.; Li, H. Y. Simulation of multiple water source ecological replenishment for Chagan Lake based on coupled hydrodynamics and water quality models. Water Sci. Technol. 2017, 17(6), 1774-1784.
Haynes, D. R.; Tro, N. J.; George, S. M.. Condensation and evaporation of water on ice surfaces. J. Phys. Chem. 1992, 96(21), 8502-8509.
Pratte, P.; van den Bergh, H.; Rossi, M. J. The kinetics of H2O vapor condensation and evaporation on different types of ice in the range 130− 210 K. J. Phys. Chem. A, 2006, 110(9), 3042-3058.
Can salinity also increase due to aquaculture activities?
[Authors’ response] The main ecological service of Chagan Lake is freshwater fish farming. Therefore, salinity is a key limiting factor for the ecological health of Chagan Lake. We mainly discuss the importance of lake salinity control with respect to the threshold of freshwater fish farming, neglecting the effect of fish on salinity. There are less literatures on the effect of freshwater fish farming on salinity increasing, but there are many literatures that lake salinization has a negative effect on the growth and reproduction of fish.
Line 362. “the ecological health of lakes (in general, right?)”. In any case if you refer irrigation districts it means that lakes receive irrigation discharge. The last part of the sentence is redundant.
[Authors’ response] We fully agree. We have deleted the last part of the sentence in line 379.
Line 366. also needs?
[Authors’ response] Have corrected in line 381.
Line 368. Was or is ? The authors should pay attention in the use of past or present.
[Authors’ response] Sorry for the mistake. We have corrected in line 384.
Line 381-384. “by the saline irrigation districts”. Why? Include a brief explanation and explain (briefly) differences in relation to other contexts.
[Authors’ response] Sorry for the confusion. Chagan Lake is located in the Songnen Plain, one of the three major saline-alkali land in the world, and the soil around irrigation districts is saline-alkali soil. However, the irrigation districts not mean the saline irrigaiton districts.
Thank you so much, again, for taking time to thoroughly review our manuscript. We greatly appreciate you for the helpful comments and suggestions.

Reviewer 3 Report
The English grammar need to be improved significantly before this manuscript can be published.
Working in an area with similar climate, I would be greatly concerned that the discharge from the irrigation districts would be declining as water conservation methods are adapted and that reduction in flow would have grave effects.
Some specific comments:
1) Fig. 1 needs to be bigger s that the relationship between the irrigation districts and the lake are easily to see.
2) The use of the subscripts a and b for the data in Table 1 needs to be better defined. As sated this reviewers had no idea what comparisons were being made.
3) Fig. 2 legend has to be expanded to describe the significance of the squares. It is eventually stated in the text. But that info should also in the figure legend.
4) The goodness of fit in Table 4 is not very impressive for salinity during the calibration phase based on the NSE values and that there are only 6 observations. It was good to see a significant increase in NSE values during validation.
5) The red points in Figure 7 were not defined
Overall these results may have limited application because the depth of the studied lake is quite shallow. This seems to be an unusual case.
Author Response
[Water] Manuscript ID: water-831030
Effects of irrigation discharge on salinity of a large freshwater lake: A case study in Chagan Lake, Northeast China
Reviewers' comments:
Reviewer3#
The English grammar need to be improved significantly before this manuscript can be published.
[Authors’ response] We have made a deal of effort to check on the English grammar and usage, and have tried our best to make our paper more readable. The final version of the manuscript has been proofread by a proficient English writer.
Working in an area with similar climate, I would be greatly concerned that the discharge from the irrigation districts would be declining as water conservation methods are adapted and that reduction in flow would have grave effects.
[Authors’ response] We would not decline the discharge from the irrigation districts using water conservation methods. The ratio of Hadashan water diversion and irrigation discharge should be changed for diluting salinity of the Chagan Lake. However, the input water of Chagan Lake remains unchanged.
Some specific comments:
1) Fig. 1 needs to be bigger s that the relationship between the irrigation districts and the lake are easily to see.
[Authors’ response] Your point is well taken. We have redrawn Figure 1 as follow:
2) The use of the subscripts a and b for the data in Table 1 needs to be better defined. As sated this reviewers had no idea what comparisons were being made.
[Authors’ response] Sorry for the confusion. We have corrected in line 1118-119 as following:
Number a and b represents significance levels of these parameters passing the 0.0001 and 0.001 significance test, respectively.
3) Fig. 2 legend has to be expanded to describe the significance of the squares. It is eventually stated in the text. But that info should also in the figure legend.
[Authors’ response] Thank you for the good suggestion. We have added the explanation in lines 156-157 as following:
Squares indicate the calculation cell.
4) The goodness of fit in Table 4 is not very impressive for salinity during the calibration phase based on the NSE values and that there are only 6 observations. It was good to see a significant increase in NSE values during validation.
[Authors’ response] 30 observation data from 6 observation stations and 5 months are used for model simulation. The NSE values of model validation have reached the standard of model operation according to Xu's study (2017)
Xu, C.; Zhang, J.; Bi, X.; Xu, Z.; He, Y.; Gin, K. Y. H. Developing an integrated 3D-hydrodynamics and emerging contaminant model for assessing water quality in a Yangtze Estuary Reservoir. Chemosphere 2017, 188, 218-230.
5) The red points in Figure 7 were not defined
[Authors’ response] Thank you for the suggestion. We have redrawn the Figure 6 and changed the legend as following:
Figure 6. The tendency of salinity in Chagan Lake from 2008 to 2018. Each boxplot illustrates the median and inter-quartile range and the whiskers indicate minimum and maximum values. The blue solid line indicates the fitting line of salinity. The black points indicate the outliers.
Overall these results may have limited application because the depth of the studied lake is quite shallow. This seems to be an unusual case.
[Authors’ response] We mainly focus on effects of irrigation discharge on salinity of the freshwater lake and we believe the methodology of this study would be suitable for other shallow lake management and application.
Thank you so much, again, for taking time to thoroughly review our manuscript. We greatly appreciate you for the helpful comments and suggestions.

Round 2
Reviewer 2 Report
The authors have globally answered all my comments and included several improvements. Nonetheless, I believe that some further work is needed particularly in what concerns text editing and revision of English.
Some examples:
Abstract:
line 20 "affect"
Line 28: The - The
Line 30: "in the summer season of the region" ? I believe I understand what you mean, but the sentence sounds a little awkward.
Line 31. western area - ...region?
Introduction
Lines 43-44. The services are prone to damaging...? The sentence needs to be rephrased. Furthermore, this single sentence does not correspond to a sufficient introduction to the theme.
Line 50. "... are becoming increasing urgent as the sources of freshwaters become scarcer..." rephrase
Lines 52-53. " A number of studies have shown that agricultural activities, which using pesticides and fertilizers,"?
Line 94. Changan Lake; aiming to:
Materials and Methods
Line 138. "Chagan Lake is to receive irrigation district from the Qian’an irrigation district in the future" . rephrase
...
Author Response
[Water] Manuscript ID: water-831030
Effects of irrigation discharge on salinity of a large freshwater lake: A case study in Chagan Lake, Northeast China
Reviewers' comments:
Reviewer2#
The authors have globally answered all my comments and included several improvements. Nonetheless, I believe that some further work is needed particularly in what concerns text editing and revision of English.
We greatly appreciate all the comments from the reviewer to help improve this manuscript. We have made a deal of effort to check on the English grammar and usage, and have tried our best to make our paper more readable in the revised manuscript. All modifications have been marked with yellow.
Some examples:
Abstract:
Line 20 "affect"
Sorry for the mistake. We have correct it in line 22.
Line 28: The - The
Sorry for the mistake. We have correct it in line 30.
Line 30: "in the summer season of the region"? I believe I understand what you mean, but the sentence sounds a little awkward.
Sorry for the confusion. We have rephrase this sentence in lines 30-32. Specifically as following:
It revealed that the mean salinity in this lake was about 0.55 psu in the summer and 0.53 psu in the winter, respectively, whereas it could reach 0.62 psu in the western part of the lake from May to October.
Line 31. western area - region?
Sorry for the mistake. We have replaced the “region” with “part” in full text. Specifically as lines 32, 37, 136, 223, 269, 271, 272, 347, 373-376, 397, 399, 412.
Introduction
Lines 43-44. The services are prone to damaging? The sentence needs to be rephrased. Furthermore, this single sentence does not correspond to a sufficient introduction to the theme.
Thanks for your good suggestion. We have rephrase this sentence in lines 45-46. Specifically as following:
Freshwater salinization, which is mainly caused by climate change and human activity, has received increasing attention due to its potential threat to the biological communities.
Line 50. " are becoming increasing urgent as the sources of freshwaters become scarcer" rephrase
Thanks for your good suggestion. We have rephrase this sentence in lines 51-53. Specifically as following:
Systematic studies on how agricultural activities affect lake salinization become urgent in the face of an increasing shortage of fresh water resources.
Lines 52-53. "A number of studies have shown that agricultural activities, which using pesticides and fertilizers,"?
Sorry for the confusion. Fertilizers are composed of various salts, so long-term and excessive application of these salt-based fertilizers will increase the salinity of the soil solution.
Kirchmann H, Pettersson S. Human urine - Chemical composition and fertilizer use efficiency[J]. Fertilizer research, 1994, 40(2):149-154.
Udawatta R. P., Motavalli P. P., Garrett H. E., Krstansky J. J. Nitrogen losses in runoff from three adjacent agricultural watersheds with claypan soils. Agriculture, Ecosystems & Environment, 2006, 117(1), 39-48.
Bouraima A.K., He B., Tian T. Runoff, nitrogen (N) and phosphorus (P) losses from purple slope cropland soil under rating fertilization in Three Gorges Region [J]. Environmental Science and Pollution Research, 2016, 23 (5), 4541–4550.
Line 94. Changan Lake; aiming to:
Sorry for the confusion. We have split this sentence into two sentences in line 98. Specifically as following:
In this study, we analyzed historical datasets collected from monitoring stations and developed an integrated hydrodynamics-salinity model to simulate spatio-temporal dynamics of salinity distributions in Chagan Lake. Our objectives are to:
Materials and Methods
Line 138. "Chagan Lake is to receive irrigation district from the Qian’an irrigation district in the future". rephrase
Sorry for the confusion. We have rephrase the sentence in line 114-116. Specifically as following:
The development plan of Songyuan irrigation district also implies that Chagan Lake will receive irrigation discharge from the Qian’an irrigation district in the future.
Thank you so much, again, for taking time to thoroughly review our manuscript. We greatly appreciate you for the helpful comments and suggestions.
Reviewer 3 Report
The authors have adequately addressed my concerns. I can recommend approve for publication, although the editors should proofread the text closely so that the published text conforms to the journal's standards for English.
Author Response
[Water] Manuscript ID: water-831030
Effects of irrigation discharge on salinity of a large freshwater lake: A case study in Chagan Lake, Northeast China
Reviewers' comments:
Reviewer3#
The authors have adequately addressed my concerns. I can recommend approve for publication, although the editors should proofread the text closely so that the published text conforms to the journal's standards for English.
We greatly appreciate all the comments from the anonymous reviewers to help improve this manuscript. Revisions have been made based on the review comments, as explained below. We have made a deal of effort to check on the English grammar and usage, and have tried our best to make our paper more readable in revised manuscript. All modifications have been marked with yellow.
Thank you so much, again, for taking time to thoroughly review our manuscript. We greatly appreciate you for the helpful comments and suggestions.